# Cortical and Subcortical Alterations and Clinical Correlates after Traumatic Brain Injury

**DOI:** 10.3390/jcm11154421

**Published:** 2022-07-29

**Authors:** Qiang Xue, Linbo Wang, Yuanyu Zhao, Wusong Tong, Jiancun Wang, Gaoyi Li, Wei Cheng, Liang Gao, Yan Dong

**Affiliations:** 1Department of Neurosurgery, Eastern Hepatobiliary Surgery Hospital, Navy Medical University, Shanghai 200433, China; smmuxq@foxmail.com (Q.X.); wc_jiancun@yeah.net (J.W.); 2Institute of Science and Technology for Brain-inspired Intelligence, Fudan University, Shanghai 210023, China; linbowang@fudan.edu.cn (L.W.); wcheng.fdu@gmail.com (W.C.); 3Department of Organ Transplantation, Changzheng Hospital, Navy Medical University, Shanghai 200070, China; zhaoyuanyu0617@163.com; 4Department of Neurosurgery, Shanghai Pudong New Area People′s Hospital, Shanghai 201299, China; tongws0599@163.com; 5Department of Neurosurgery, People′s Hospital of Putuo District, Tongji University School of Medicine, Shanghai 200061, China; ligaoyitongji@126.com; 6Department of Neurosurgery, Shanghai Tenth People′s Hospital, Tongji University School of Medicine, Shanghai 200072, China

**Keywords:** traumatic brain injury, gray matter volume, white matter track, prognosis

## Abstract

**Background**: Traumatic brain injury (TBI) often results in persistent cognitive impairment and psychiatric symptoms, while lesion location and severity are not consistent with its clinical complaints. Previous studies found cognitive deficits and psychiatric disorders following TBI are considered to be associated with prefrontal and medial temporal lobe lesions, however, the location and extent of contusions often cannot fully explain the patient′s impairments. Thus, we try to find the structural changes of gray matter (GM) and white matter (WM), clarify their correlation with psychiatric symptoms and memory following TBI, and determine the brain regions that primary correlate with clinical measurements. **Methods**: Overall, 32 TBI individuals and 23 healthy controls were recruited in the study. Cognitive impairment and psychiatric symptoms were examined by Mini-Mental State Examination (MMSE), Hospital Anxiety and Depression Scale (HADS), and Wechsler Memory Scale-Chinese Revision (WMS-CR). All MRI data were scanned using a Siemens Prisma 3.0 Tesla MRI system. T1 MRI data and diffusion tensor imaging (DTI) data were processed to analyze GM volume and WM microstructure separately. **Results**: In the present study, TBI patients underwent widespread decrease of GM volume in both cortical and subcortical regions. Among these regions, four brain areas including the left inferior temporal gyrus and medial temporal lobe, supplementary motor area, thalamus, and anterior cingulate cortex (ACC) were highly implicated in the post-traumatic cognitive impairment and psychiatric complaints. TBI patients also underwent changes of WM microstructure, involving decreased fractional anisotropy (FA) value in widespread WM tracts and increased mean diffusivity (MD) value in the forceps minor. The changes of WM microstructure were significantly correlated with the decrease of GM volume. **Conclusions**: TBI causes widespread cortical and subcortical alterations including a reduction in GM volume and change in WM microstructure related to clinical manifestation. Lesions in temporal lobe may lead to more serious cognitive and emotional dysfunction, which should attract our high clinical attention.

## 1. Introduction

Traumatic brain injury (TBI) is the leading cause of death and disability in individuals below age 45, which typically caused by motor vehicle crashes, falls, contact sports, or assaults. TBI often results in persistent cognitive impairment and psychiatric symptoms, including memory deficit, anxiety, and depression, which exert a negative impact on quality of life and rehabilitation process [1,2,3]. Due to the significant heterogeneity in the clinical presentation and neuropathological across TBI patients, it is challenging to predict the risk of comorbid psychiatric disorders and cognitive impairment [4]. Previous studies found that TBI severity is not correlated with neuropsychiatric outcome [5,6]. Meanwhile, previous findings for the correlation between lesion location and cognitive function and psychiatric complaints were inconsistent [7].

Although TBI is heterogeneous in the cause and intensity of impact, it still exhibits a featured pattern of anatomical injuries. The impact typically results in contusions involving the basal and polar regions of the frontal and temporal lobes [8,9]. Besides focal brain injuries, diffuse axonal injuries (DAI) occur after TBI, which frequently affect the frontal and temporal WM, corpus callosum, and brainstem [10]. Traditionally, cognitive deficits and psychiatric disorders following TBI are considered to be associated with prefrontal and medial temporal lobe lesions, however the location and extent of these contusions often cannot fully explain the patient′s impairments [11,12,13,14]. During the acute and subacute stages of TBI, secondary damages including inflammation, apoptosis, excitotoxicity, and prolonged hypo-perfusion result in progressive and widespread white matter (WM) atrophy and gray matter (GM) volume loss across large areas of the brain spanning most of the cortex and subcortical areas over time [14,15,16]. These abnormalities in the parietal and occipital lobes and subcortical regions including basal ganglions and thalamus are closely related to post-traumatic cognitive functions and psychiatric symptoms [14,17,18]. Due to the heterogeneity inherent to study design and various pipelines of data pre-processing, parcellation, and analysis among the previous studies, the topographical distribution of morphometric changes and their clinical associations related to post-traumatic psychiatric symptoms and memory function remains inconsistent.

In the present study, we assessed the cortical and subcortical GM and WM damage and its relation to post-traumatic neuropsychological measurements of anxiety and depressive symptoms and memory function. We then investigated the relationships between disruption of WM microstructure and the decrease in GM volume. We aimed to obtain a comprehensive understanding of the relationship between the structural alterations and psychiatric symptoms and memory following TBI and determine the brain regions correlated with clinical measurements.

## 2. Materials and Methods

### 2.1. Participants

Overall, 32 patients with TBI were recruited from the Shanghai Pudong New Area People’s Hospital. These patients were older than 18 years and with first-ever TBI and positive finding on cranial CT scans on admission. They were excluded from the study if suffering neurological or psychiatric disorders prior to TBI. The initial evaluation of severity of TBI took place within the first 24 h after injury during hospitalization based on Glasgow Coma Scale (GCS), which classifies TBI into three categories as mild (GCS 13–15), moderate (GCS 9–12), and severe (GCS 3–8). Twenty-three healthy participants without history of TBI, neurological, and psychiatric disorders were matched for age, gender, and education. The ethics committee of the Pudong New Area People’s Hospital approved the study. Informed written consent for study participation was obtained from all patients and healthy controls.

### 2.2. Cognitive Functional Assessment and Neuropsychological Assessment

The global cognition was assessed with the Mini-Mental State Examination (MMSE), which includes items measuring orientation, attention, memory, language, and visual/spatial skills. MMSE scores range from 0 to 30, with a higher score indicating better cognitive performance. The Hospital Anxiety and Depression Scale (HADS), which is a brief self-assessment questionnaire measuring severity of emotional disorder and has been validated in TBI populations, was employed to evaluate the anxiety and depressive symptoms. The anxiety and depression subscales have seven items respectively. Wechsler Memory Scale-Chinese Revision (WMS-CR) picture, recognition, associative learning, comprehension memory, and digit span were administered to evaluate multiple categories of memory capacity. The sum of five subscales was calculated to reflect general memory function.

### 2.3. Image Acquisition

All MRI data were collected using a Siemens Prisma 3.0 Tesla MRI system (Prisma, Siemens, Erlangen, Germany) equipped with a 20-channel head coil. Participants assumed a supine position in the MRI scanner with cushions to restrict the mobility of their heads, thus minimizing the head motion. During rs-fMRI scanning, participants were guided to stay awake with their eyes closed without thinking about anything in particular. Structural images were acquired using a high-resolution T1-weighted MPRAGE sequence with 192 sagittal slices, TR/TE = 2530/2.98 ms, flip angle = 7°, FOV = 256 × 256 mm, matrix size = 256 × 256, voxel size = 1 × 1 × 1 mm^3^, which facilitated the localization and co-registration of functional data. In addition, transverse turbo-spin-echo T2-weighted images for lesion localization were obtained with 30 axial slices, slice thickness = 5 mm, TR/TE = 6000/95 ms, flip angle = 120°, FOV = 220 × 220 mm, matrix size = 320 × 320, voxel size = 0.34 × 0.34 × 5 mm^3^. Diffusion tensor images (DTI) were acquired using an echo planar imaging (EPI) sequence (30 gradient directions, 1 baseline (b = 0) image, b = 1000 s/mm^2^, TR = 10,100 ms, TE = 92 ms, FOV = 256 × 256 mm, 75 axial slices, voxel size = 2.0 × 2.0 × 2.0 mm^3^.

### 2.4. T1 MRI Data Processing and Analysis

The T1-weighted MRI images were preprocessed by using the CAT12 (Computational Anatomy Toolbox; http://dbm.neuro.uni-jena.de/cat12/; accessed on 1 June 2021) for grey matter extraction, which is an extension of SPM12 (Statistical Parametric Mapping) to provide computational anatomy. Images were segmented into GM, WM, and cerebrospinal fluid (CSF), and normalized to a standard template (Montreal Neurological Institute). Raw images of lower quality (CAT image quality rating <75%) were excluded. Cortical maps were smoothed using an 8-mm full width at half maximum kernel, prior to building the statistical model. After preprocessing, the Brainnetome Atlas was used to extract regional grey matter volume by averaging voxel GM within each regions of interest (ROI). Based on the Brainnetome Atlas, GM was segmented into 246 ROI.

### 2.5. Diffusion Tensor Imaging(DTI) Data Preprocessing and Analysis

The DTI data were preprocessed by using the FMRIB Diffusion Toolbox (FSL, FMRIB, Oxford, UK). Briefly, after correcting for the eddy-current effect and brain tissue extraction, the diffusion tensor model was fit to extract DTI measures. The output yielded voxel-wise maps of fractional anisotropy (FA), mean diffusivity (MD), radial diffusivity (RD), and axial diffusivity (AD). Next, DTI data from each participant were registered to a standard space (Montreal Neurological Institute, NMI, ICBM-152). To obtain a comprehensive WM segmentation, WM tracts were defined using JHU_ICBM_tracts_maxprob_thr25 atlas [19]. Finally, mean FA, MD, RD, and AD values were computed in each WM ROI in standard space for each participant.

### 2.6. Correlation Analysis and Statistical Analyses

Statistical analyses were conducted using Matlab 2018b. Continuous variables were described using means and standard deviations, and categorical variables were summarized using frequencies. The normality distribution of continuous data was verified with a one-sample Kolmogorov-Smirnov Test. Pearson correlation was used to investigate the relationship between the structural measures and clinical scores. Age, gender, and educational level were regressed out before correlation analysis. The independent *t*-test were applied to perform the group comparisons for continuous demographic and clinical variables. The categorical demographic was computed using a Chi-square test. Statistical significance was set at *p* < 0.05 (false discovery rate [FDR] corrected). Missing data were not included in all analysis.

## 3. Results

### 3.1. Demographic and Clinical Characteristics

Thirty-two TBI patients and 23 healthy controls were included in the present study, with the demographic and clinical parameters shown in Table 1. Overall, 31.25% of patients had moderate–severe and 68.75% mild TBI. Average time since injury was 8.47 months. There were no significant differences between the TBI group and healthy controls in terms of gender (*p* = 0.949), age (*p* = 0.422) or education years (*p* = 0.756). Relative to healthy controls, TBI patients had lower level of MMSE scores. Moreover, they performed worse on memory function tests and presented more anxiety and depressive symptoms.

### 3.2. Reduced GM Volumes in Patients with Traumatic Brain Injury

As shown in Figure 1A, focal lesions were mainly present in bilateral orbitofrontal and temporal cortical regions. Compared to healthy controls, TBI patients underwent widespread decrease of GM volume in the bilateral frontal and temporal gyrus, left cingulate gyrus, and right insular lobe, particularly in the right orbitofrontal, left inferior, and middle temporal lobe, and subgenual anterior cingulate cortex (Figure 1B, Table 2). In subcortical regions, decreased GM volumes were observed in the bilateral amygdala, right hippocampus, left nucleus accumbens, and bilateral rostral temporal thalamus.

### 3.3. Correlations between GM Volume and Clinical Parameters in Patients with Traumatic Brain Injury

Although extensive atrophy was observed in the cortical and subcortical structures, only a small set of brain regions correlate with the clinical parameters, as shown in Table 3. There were significant correlations (*p* < 0.05 uncorrected) between MMSE scores and GM volumes in the left inferior temporal gyrus extending to the left fusiform gyrus and middle temporal gyrus, bilateral hippocampus, left anterior cingulate cortex (ACC), and left thalamus. Specifically, the total memory scores were associated with the GM volume of the left thalamus, right middle frontal gyrus, and insular lobe. Analysis of the anxiety and depressive symptoms showed that the GM volume of the supplementary motor area was correlated to anxiety and depressive symptoms. In addition, the anxiety symptoms were also significantly associated with the decreased GM volume of the left hippocampus. In brief, four brain areas–the left inferior temporal gyrus and medial temporal lobe, supplementary motor area, thalamus, and ACC–were highly implicated in the post-traumatic cognitive impairment and psychiatric complaints.

### 3.4. WM Microstructure Alterations in Patients with Traumatic Brain Injury

Compared with the healthy controls, TBI patients showed a significantly decreased FA value in widespread WM tracts, including the left inferior fronto-occipital fasciculus (IFOF), left superior longitudinal fasciculus (SLF), bilateral uncinate fasciculus (UF), forceps major, and forceps minor (Figure 2, Table 4). Mean diffusivity of WM regions showed significant difference in the forceps minor.

As UF and IFOF traverse the temporal lobe, which is closely related to the post-traumatic cognitive function and psychiatric symptoms, we further explored the relationship between these two fiber tracks and the GM structures of the temporal gyrus and medial temporal lobe. GM volumes were significantly related to the FA of left UF and left IFOF in multiple regions of left temporal gyrus and left hippocampus (Table 5). No statistically significant association was found between SLF and the GM volumes of the supplementary motor area.

## 4. Discussion

In this work, we systemically investigated differences in whole-brain GM and WM between participants with TBI and healthy controls and explored their relationships with clinical measurements. We first demonstrated TBI patients underwent widespread decrease of GM volume in both cortical regions and subcortical regions. Among these regions, four brain areas including left inferior temporal gyrus and medial temporal lobe, supplementary motor area, thalamus, and ACC were highly implicated in the post-traumatic cognitive impairment and psychiatric complaints. We then found WM microstructure was disrupted in TBI patients, involving a decreased FA value in widespread WM tracts including the left IFOF, left SLF, bilateral UF, forceps major, and forceps minor, with an increased MD value in the forceps minor. Finally, we explored the consistency of structural damage in gray and WM, showing GM volumes were significantly related to the FA of left UF and left IFOF in multiple regions of left temporal gyrus and left hippocampus.

About 1.7 million people in the United States develop TBI each year, and more than 50,000 people have severe cognitive impairment [20]. In China, TBI occupies second place in the incidence of systemic trauma and first place in the fatality and disability rate [21]. Most TBI patients are young and middle-aged patients, and may suffer from long-time loss of living and working ability. The primary injury of TBI may not be serious, but due to changes in the local pathological environment after trauma, it is easy to induce diffuse axonal injury, which indicates damage to axons and surrounding fibers [22]. Patients gradually develop affective and cognitive dysfunction. Therefore, it is of great clinical significance to study the changes of brain microstructure after TBI and its relationship with cognitive–emotional function.

In accordance with previous reports, TBI patients presented common abnormalities of brain morphology despite heterogeneity in injury severity and mechanisms, extent of focal insults, and time since injury [23,24,25]. Despite widespread atrophy across the cortical and subcortical regions, only a relatively small subset of this pattern of damage–mainly in the left inferior temporal gyrus and medial temporal lobe, supplementary motor area, thalamus, and ACC–were highly implicated in the post-traumatic cognitive impairment and psychiatric complaints. Among 32 TBI patients, those with temporal lobe injury had more severe symptoms than those with frontal lobe injury, especially those with lesions on the inferior temporal gyrus. Moreover, TBI causes extensive changes in the WM microstructure, including the left IFOF, left SLF, bilateral UF, forceps major, and forceps minor. GM volumes in multiple subregions of left temporal lobe and left hippocampus were significantly related to the FA of left UF and left IFOF. This provides some inspiration for our clinical work. Although the frontal lobe and temporal lobe are most likely to be injured in TBI, differences exist in their clinical manifestation and outcome [26,27]. Injury of the temporal lobe should be paid more attention; due to its anatomical position in the brain, the temporal lobe contains a large number of association fibers and commissural fibers. It has a complex integration effect on the frontal lobe, occipital lobe, and the sensory motor areas of parietal lobe, and plays a coordinating role between the anterior–posterior and left–right brain. The completion of brain function depends on the multi-synaptic information transmission of nerve fibers, and damage to the temporal lobe interrupts or affects the transmission of this information, resulting in cognitive and emotional dysfunction. Therefore, even small lesions in the temporal lobe should attract our high clinical attention. Early clinical monitoring of cognitive and emotional functions in patients with craniocerebral injury, and timely, comprehensive cognitive rehabilitation intervention is of great significance to prevent further functional decline and improve the quality of life of patients.

In our research, we demonstrated a widespread GM volume reduction in both cortical regions and subcortical regions. In addition to the reduction in GM volume at the immediate injury lesion, some deep structural GM volumes were also reduced and correlated with the patient′s clinical presentation. Changes in the local metabolic environment and the occurrence of DAI may be responsible for this [28,29,30]. Eventually, patients have axonal and fiber damage, and gradually develop cognitive impairment and altered affective function. We also calculated the FA and MD values of the WM fiber tracts, and found a significantly decreased FA value in widespread WM tracts including the left IFOF, left SLF, bilateral UF, forceps major, and forceps minor, with a significantly increased MD value in the forceps minor. Among these fiber tracks, the forceps minor and forceps major are the interhemispheric fibers of the frontal cortex and occipital cortex, respectively, and the UF, IFOF, and SLF are intrahemispheric association fibers. The UF and IFOF connect the temporal lobe with orbital and polar frontal cortex, and the SLF links the frontal lobe with parietal lobe. This means that although the damage mostly occurs in the cerebral cortex in TBI patients, irreversible structural damage to WM fiber tracts still occurs, and both intrahemispheric associative fibers and interhemispheric fibers will break, leading to the appearance of symptoms in patients.

The human brain has an overall leftward posterior and rightward anterior asymmetry, which may help to provide cognitive advantages and solve spatial constraints [31,32]. Existing studies have confirmed that there is brain asymmetry in normal aging or neuropsychiatric and neurodegenerative diseases [33,34]. Under pathological conditions, the left hemisphere may be mainly affected [35]; we found such leftward lateralization in our research. Damage to the integrity of the left IFOF and left SLF is significantly stronger than that of the right side, which is also consistent with the reduction of GM volume of left temporal gyrus and left hippocampus. This can be explained as TBI promotes the degradation of WM in the non-dominant hemisphere or leads to the transformation of structure to the dominant hemisphere. However, any inference about the change direction of symmetry after TBI is complex and needs further exploration.

Early parcellation efforts aimed at defining regional boundaries using limited samples, including the widely used Brodmann atlas and automated anatomical labeling (AAL) atlas [36,37]. The Brainnetome Atlas is a connectivity-based parcellation of the brain, which establishes a priori, biologically valid brain parcellation scheme of the entire cortical and subcortical GM into sub-regions showing a coherent pattern of anatomical connections and provides a new framework for human brain research and in particular connectome analysis [38,39,40,41]. Thus, in this research we used The Brainnetome Atlas to more accurately describe the locations of the activation or connectivity in the brain.

We also analyzed the limitations of our research for further improvement. First, the population enrolled in this study was relatively small and consisted of Chinese only. With the different brain injured regions of patients, the degree of injury is not consistent, which may lead to high heterogeneity among patients and an impact on the results. Secondly, patients are often accompanied with cognitive impairment and emotional dysfunction; these confounding factors will add to the difficulty of analysis of drawing conclusions on the correlation between changes in gray and WM structure and cognitive and affective dysfunction. Thirdly, the HADS scale for emotional dysfunction is a self-reported questionnaire. It is possible that TBI patients with cognitive impairment tend to overestimate or underestimate their mood problems, which may reduce the credibility of the conclusion. Finally, as a retrospective cross-sectional study, it is prone to produce selection bias and recall bias, which affects the precision of the outcome.

In conclusion, we shed light on differences in whole-brain GM and WM maps and explored their clinical significance. Briefly, four brain regions, including the left inferior temporal gyrus and medial temporal lobe, supplementary motor area, thalamus, and ACC, correlated to cognitive performance and psychiatric complaints following TBI, and injury of the temporal lobe should be paid more attention.

## Figures and Tables

**Figure 1 jcm-11-04421-f001:**
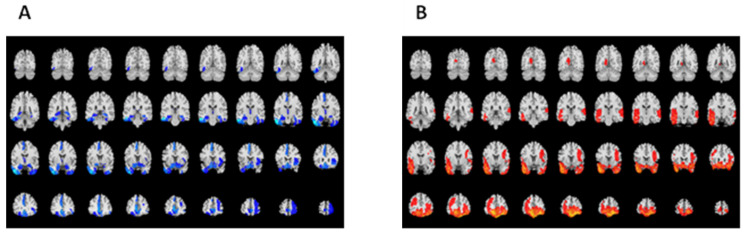
Lesion map following TBI (**A**) and differences in GM volumes between TBI participants and healthy controls (**B**).

**Figure 2 jcm-11-04421-f002:**
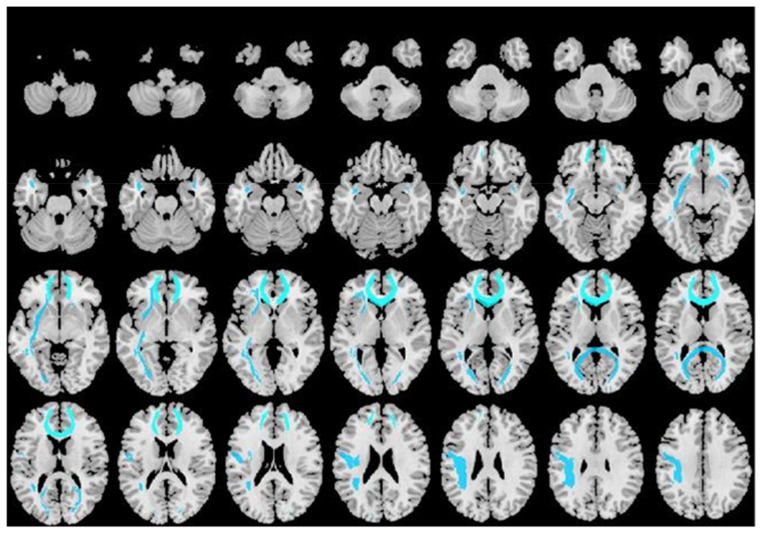
Voxel-wise Tract-Based Spatial Statistics differences in FA metrics between group.

**Table 1 jcm-11-04421-t001:** The demographic and clinical characteristics in TBI patients and healthy controls.

	TBI (*n* = 32)	Healthy Controls (*n* = 23)	*p* Value
Age, mean (SD), y	35.59 (10.64)	33.35 (9.42)	0.422
Male, No. (%)	22 (68.35%)	16 (69.57%)	0.949
Educational lever, mean (SD), y	9.22 (4.16)	9.57 (3.89)	0.756
Time since injury, mean (SD), m	8.41 (7.02)	NA	NA
GCS, No. (%)			
13–15	22 (68.75%)	NA	NA
9–12	6 (12.89%)	NA	NA
3–8	4 (12.50%)	NA	NA
MMSE, mean (SD)	27.281 (2.57)	29.26 (0.92)	0.001
HADS anxiety, mean (SD)	8.63 (4.65)	4.22 (2.43)	<0.001
HADS depression, mean (SD)	7.97 (5.96)	3.09 (2.25)	<0.001
Memory, mean (SD)	40.31 (13.54)	52.04 (7.10)	<0.001

TBI: traumatic brain injury; GCS: Glasgow coma scale; MMSE: mini mental state examination; HADS: hospital anxiety and depression scale; NA: not applicable.

**Table 2 jcm-11-04421-t002:** Brain regions with statistical difference between TBI patients and healthy controls.

Regions ^#^	TBI (*n* = 32) Mean (*±*SD), mm^3^	HC (*n* = 23) Mean (*±*SD), mm^3^	% of Volumetric Decreases	*p* Value
SFG_L_7_1	679.38 *±* 95.23	763.34 *±* 80.17	11.00%	0.019
SFG_L_7_5	688.43 *±* 90.55	762.78 *±* 79.24	9.75%	0.033
SFG_L_7_6	617.15 *±* 104.01	704.67 *±* 91.78	12.42%	0.033
MFG_R_7_3	770.74 *±* 175.19	914.94 *±* 132.80	15.76%	0.041
MFG_R_7_7	724.51 *±* 155.40	851.19 *±* 112.29	14.88%	0.045
IFG_R_6_5	522.33 *±* 75.282	601.15 *±* 101.43	13.11%	0.034
OrG_L_6_1	415.58 *±* 88.91	487.35 *±* 65.21	14.73%	0.048
OrG_R_6_1	549.00 *±* 129.35	677.97 *±* 98.88	19.02%	0.017
OrG_L_6_3	760.58 *±* 144.75	886.59 *±* 110.28	14.21%	0.034
OrG_L_6_5	907.44 *±* 148.74	1056.28 *±* 137.44	14.09%	0.017
OrG_R_6_5	755.51 *±* 134.16	879.01 *±* 112.78	14.05%	0.033
OrG_R_6_6	412.28 *±* 67.12	472.76 *±* 56.05	12.79%	0.034
STG_L_6_1	632.45 *±* 129.23	727.36 *±* 90.08	13.05%	0.048
MTG_L_4_2	692.60 *±* 151.93	835.50 *±* 126.73	17.10%	0.017
ITG_L_7_1	252.28 *±* 45.31	300.55 *±* 45.66	16.06%	0.017
ITG_L_7_3	461.55 *±* 84.69	568.62 *±* 81.82	18.83%	0.005
ITG_R_7_3	410.09 *±* 73.29	474.31 *±* 55.33	13.54%	0.034
ITG_L_7_4	431.60 *±* 89.55	551.51 *±* 82.59	21.74%	<0.001
ITG_R_7_4	480.15 *±* 80.21	553.43 *±* 82.71	13.24%	0.035
ITG_L_7_7	497.54 *±* 89.74	586.11 *±* 92.01	15.11%	0.017
FuG_L_3_1	997.68 *±* 149.18	1154.11 *±* 146.99	13.55%	0.017
FuG_R_3_1	1111.18 *±* 164.05	1244.75 *±* 145.85	10.73%	0.047
FuG_L_3_3	948.99 *±* 136.76	1061.95 *±* 148.23	10.64%	0.050
PhG_L_6_5	115.27 *±* 17.03	130.30 *±* 18.02	11.53%	0.039
INS_R_6_2	246.82 *±* 31.88	276.64 *±* 37.96	10.78%	0.035
INS_R_6_3	285.86 *±* 41.35	323.85 *±* 54.10	11.73%	0.050
CG_L_7_3	470.88 *±* 88.78	554.36 *±* 66.22	15.06%	0.017
CG_L_7_7	633.52 *±* 147.04	794.16 *±* 118.78	20.23%	0.006
Amyg_L_2_1	185.75 *±* 21.72	207.61 *±* 26.65	10.53%	0.034
Amyg_R_2_1	267.19 *±* 32.58	297.09 *±* 38.82	10.07%	0.049
Amyg_L_2_2	93.16 *±* 10.40	103.25 *±* 12.28	9.77%	0.033
Amyg_R_2_2	140.75 *±* 15.64	156.67 *±* 18.30	10.16%	0.028
Hipp_L_2_1	666.67 *±* 80.30	737.97 *±* 80.84	9.66%	0.034
Hipp_L_2_2	485.14 *±* 67.36	541.11 *±* 64.92	10.34%	0.050
Hipp_R_2_2	566.44 *±* 75.22	633.31 *±* 60.04	10.56%	0.034
BG_L_6_3	368.88 *±* 49.71	411.61 *±* 48.00	10.38%	0.033
BG_R_6_3	456.63 *±* 64.02	507.36 *±* 57.34	10.00%	0.050
Tha_L_8_4	174.26 *±* 30.13	198.40 *±* 24.08	12.17%	0.034
Tha_R_8_4	185.47 *±* 33.66	213.33 *±* 28.92	13.06%	0.034

^#^ Brainnetome atlas; TBI: traumatic brain injury; HC: healthy controls.

**Table 3 jcm-11-04421-t003:** Significant correlations between the GM volumes and clinical parameters in TBI patients.

Regions ^#^	MMSE		Memory		HADS-A		HADS-D	
	r_Value	*p*_Value	r_Value	*p*_Value	r_Value	*p*_Value	r_Value	*p*_Value
SFG_L_7_1	NS	NS	NS	NS	NS	NS	0.36	0.04
SFG_L_7_5	NS	NS	NS	NS	0.37	0.03	NS	NS
MFG_R_7_3	NS	NS	−0.34	0.05	NS	NS	NS	NS
MTG_L_4_2	0.45	0.01	NS	NS	NS	NS	NS	NS
ITG_L_7_1	0.37	0.03	NS	NS	NS	NS	NS	NS
ITG_L_7_4	0.49	<0.01	NS	NS	NS	NS	NS	NS
ITG_L_7_7	0.36	0.04	NS	NS	NS	NS	NS	NS
FuG_L_3_1	0.45	0.01	NS	NS	NS	NS	NS	NS
FuG_L_3_3	0.58	<0.01	NS	NS	NS	NS	NS	NS
INS_R_6_2	NS	NS	−0.45	0.01	NS	NS	NS	NS
CG_L_7_3	0.40	0.02	NS	NS	NS	NS	NS	NS
Hipp_L_2_2	0.55	<0.01	NS	NS	0.38	0.03	NS	NS
Hipp_R_2_2	0.44	0.01	NS	NS	NS	NS	NS	NS
Tha_L_8_4	0.49	<0.01	0.51	<0.01	NS	NS	NS	NS

^#^ Brainnetome atlas; NS: not significant MMSE: mini mental state examination; HADS-A: hospital anxiety and depression scale: anxiety; HADS-D: hospital anxiety and depression scale: depression.

**Table 4 jcm-11-04421-t004:** Significant outcome of WM integrity between TBI patients and healthy controls.

Regions ^#^	Fractional Anisotropy			Mean Diffusivity		
	TBI (*n* = 32) Mean (±SD)	HC (*n* = 23) Mean (±SD)	*p*-Value	TBI (*n* = 32) Mean (±SD)	HC (*n* = 23) Mean (±SD)	*p*-Value
Forceps.major	0.68 ± 0.02	0.70 ± 0.02	0.03	NS	NS	NS
Forceps.minor	0.53 ± 0.03	0.56 ± 0.03	0.01	0.00077 ± 0.000036	0.00073 ± 0.000043	0.04
Inferior.fronto-occipital.fasciculus.L	0.51 ± 0.03	0.53 ± 0.03	0.04	NS	NS	NS
Superior.longitudinal.fasciculus.L	0.48 ± 0.03	0.49 ± 0.03	0.04	NS	NS	NS
Uncinate.fasciculus.L	0.48 ± 0.05	0.52 ± 0.03	0.03	NS	NS	NS
Uncinate.fasciculus.R	0.52 ± 0.05	0.55 ± 0.03	0.04	NS	NS	NS

^#^ JHU White-Matter Tractography Atlas; NS: not significant; TBI: traumatic brain injury; HC: healthy controls.

**Table 5 jcm-11-04421-t005:** The relationship between fractional anisotropy and GM volume in the temporal gyrus and medial temporal lobe of TBI Subjects.

	Left Uncinate Fasciculus		Left Inferior Fronto-Occipital Fasciculus	
Regions ^#^	r	*p*	r	*p*
STG_L_6_1	0.358	0.041	NS	NS
STG_L_6_2	−0.439	0.011	−0.399	0.021
STG_L_6_5	0.435	0.011	NS	NS
STG_L_6_6	0.347	0.048	NS	NS
MTG_L_4_1	0.391	0.025	NS	NS
MTG_L_4_2	0.555	<0.001	NS	NS
MTG_L_4_3	0.491	0.004	0.427	0.013
ITG_L_7_1	0.504	0.003	0.573	<0.001
ITG_L_7_3	0.398	0.022	NS	NS
ITG_L_7_4	0.397	0.022	NS	NS
ITG_L_7_5	0.403	0.020	NS	NS
ITG_L_7_6	0.542	0.001	0.530	0.002
FuG_L_3_1	0.471	0.006	NS	NS
FuG_L_3_3	NS	NS	0.379	0.029
PhG_L_6_1	0.422	0.014	NS	NS
PhG_L_6_2	NS	NS	0.387	0.026
PhG_L_6_4	0.464	0.007	0.359	0.040
PhG_L_6_5	0.598	<0.001	NS	NS
Hipp_L_2_1	0.559	<0.001	NS	NS
Hipp_L_2_1	0.512	0.002	NS	NS

^#^ Brainnetome atlas; NS: not significant.

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
