# Peer review of "Cortical and Subcortical Alterations and Clinical Correlates after Traumatic Brain Injury"

_jcm, 2022, doi:10.3390/jcm11154421_

Round 1
Reviewer 1 Report
Dear Authors,
I read your work entitled “Cortical and subcortical alterations and clinical correlates after traumatic brain injury” and here I have my recommendations:
Introduction
The introduction is well organised but there is a need to have a sounder rational behind this study.
Also, correct this
“… Besides focal brain injuries, diffuse axonal injuries (DAI) occur after TBI, which frequently affect the frontal and temporal WM, corpus callosum, and brainstem [10]….”
Have the acronym WM described it here and not in line 63 first “…. prolonged hypo-perfusion result in progressive and widespread white matter (WM) atrophy and gray matter(GM) volume loss across large areas of the brain spanning…. ”
Methods
All methods section has no citations for example in the following paragraphs there are not citations of the scales and the procedures:
“Participants
The initial evaluation of severity of TBI took place within the first 24 hours after injury during hospitalization based on Glasgow Coma Scale (GCS), which classifies TBI into three categories as mild (GCS 13-15), moderate (GCS 9-12) and severe (GCS 3-8). Twenty-three healthy participants without history of TBI, neurological and psychiatric disorders were matched for age, gender and education. The ethics committee of the Pudong New Area People's Hospital approved the study. Informed written consent for study participation was obtained from all patients and healthy controls….”
Cognitive Functional Assessment and Neuropsychological Assessment
“…. The global cognition was assessed with the Mini-Mental State Examination (MMSE), which includes items measuring orientation, attention, memory, language, and visualspatial skills. MMSE scores range from 0 to 30, with a higher score indicating better cognitive performance. The Hospital Anxiety and Depression Scale (HADS), which is a brief self-assessment questionnaire measuring severity of emotional disorder and has been validated in TBI populations, was employed to evaluate the anxiety and depressive symptoms. The anxiety and depression subscales have seven items respectively. Wechsler Memory Scale-Chinese Revision (WMS-CR) picture, recognition, associative learning, comprehension memory and digit span were administered to evaluate multiple categories of memory capacity. The sum of five subscales was calculated to reflect general memory function….”
Results
§ Table 2 must be organised better informing the reader what are those acronyms are and which p-values are statistically significant different maybe a footnote under the table would help.
§ Table 3 must be organised better informing the reader what are those acronyms are and which correlations are statistically significant different maybe a footnote under the table as well would help.
§ Table 4 must have a sign which values are statistically significant different.
§ Table 5 must be organised better informing the reader what are those acronyms are and which correlations are statistically significant different maybe a footnote under the table as well would help.
Thank you.
Author Response
Thank you for your advice sincerely. Please see the attachment.

Reviewer 2 Report
The authors provided a summary of a clinically relevant project included persons with and without TBI to compare MRI brain imaging studies with standardized clinical cognitive assessments for memory, anxiety/depression and emotional dysfunction. Using digital brain atlas references, white (WM) and grey matter (GM) volumes four specific brain regions were found to be significantly correlated to cognitive and emotional assessment scores. This work is of great interest to both clinicians and researchers to advance diagnostic and treatment protocols. Several imaging study images added to the quality of information shared. However there were some minor important items which should be addressed:
1. There were several minor grammatical edits suggested through which should be simple to correct. (Please see attached document for edits)
2. Several tables could be slightly modified to improve understanding.
3. One table #5 appears to be incomplete and missing r-values to associate with p-values.
4. Last of all, authors appears to over step the scope of study findings with some conclusions in the discussion. A slight modification of language has been suggested.
Overall, enjoyed reading about the authors work and appreciate their contribution to the research.

Author Response

(The authors gave the same response as above.)

Round 2
Reviewer 1 Report
Dear Authors,
I read your work entitled “Cortical and subcortical alterations and clinical correlates after traumatic brain injury” and here I enclose my final recommendations to you:
A) Tables. Please report p-values properly for example in Table 2.
Table 2 Brain regions with statistical difference between TBI patients and healthy controls
|
|
|
Regions# |
TBI (n = 32) |
HC (n = 23) |
% of volumetric |
P value |
|
|
|
|
Mean (±SD), mm3 |
Mean (±SD), mm3 |
decreases |
|
|
|
|
SFG_L_7_1 |
679.38±95.23 |
763.34±80.17 |
11.00% |
<0.05 |
|
|
|
|
|
|
|
|
|
|
|
SFG_L_7_5 |
688.43±90.55 |
762.78±79.24 |
9.75% |
or =0.032 |
Please apply to all Tables besides Table 1
Thank you.
Author Response
Thank you for your kind suggestion. We have rechecked the p-values and revised them more properly among all tables follow your suggestion. Thank you very much for raising this point.